# A Cellular Assay for Spike/ACE2 Fusion: Quantification of Fusion-Inhibitory Antibodies after COVID-19 and Vaccination

**DOI:** 10.3390/v14102118

**Published:** 2022-09-25

**Authors:** Fabien Abdul, Pascale Ribaux, Aurélie Caillon, Astrid Malézieux-Picard, Virginie Prendki, Nathalie Vernaz, Nikolay Zhukovsky, Flavien Delhaes, Karl-Heinz Krause, Olivier Preynat-Seauve

**Affiliations:** 1Department of Microbiology and Molecular Medicine, Faculty of Medicine, University of Geneva, 1 Rue Michel Servet, 1211 Geneva, Switzerland; 2Department of Pathology and Immunology, Faculty of Medicine, University of Geneva, 1 Rue Michel Servet, 1211 Geneva, Switzerland; 3Division of Internal Medicine for the Aged, Department of Rehabilitation and Geriatrics, Geneva University Hospitals, Chemin du Pont Bochet 3, 1226 Thônex, Switzerland; 4Division of Clinical Pharmacology and Toxicology, Geneva University Hospitals, Rue Gabrielle-Perret-Gentil 4, 1205 Geneva, Switzerland; 5Medical Directorate, Geneva University Hospitals, Rue Gabrielle-Perret-Gentil 4, 1205 Geneva, Switzerland; 6Neurix SA, 64 Avenue de la Roseraie, 1205 Geneva, Switzerland; 7Department of Medicine, Faculty of Medicine, University of Geneva, 1 Rue Michel Servet, 1211 Geneva, Switzerland

**Keywords:** SARS-CoV-2, cell fusion, dual-split luciferase, neutralizing antibody

## Abstract

Not all antibodies against SARS-CoV-2 inhibit viral entry, and hence, infection. Neutralizing antibodies are more likely to reflect real immunity; however, certain tests investigate protein/protein interaction rather than the fusion event. Viral and pseudoviral entry assays detect functionally active antibodies but are limited by biosafety and standardization issues. We have developed a Spike/ACE2-dependent fusion assay, based on a split luciferase. Hela cells stably transduced with Spike and a large fragment of luciferase were co-cultured with Hela cells transduced with ACE2 and the complementary small fragment of luciferase. Cell fusion occurred rapidly allowing the measurement of luminescence. Light emission was abolished in the absence of Spike and reduced in the presence of proteases. Sera from COVID-19-negative, non-vaccinated individuals or from patients at the moment of first symptoms did not lead to a significant reduction of fusion. Sera from COVID-19-positive patients as well as from vaccinated individuals reduced the fusion. This assay was more correlated to pseudotyped-based entry assay rather than serology or competitive ELISA. In conclusion, we report a new method measuring fusion-inhibitory antibodies in serum, combining the advantage of a complete Spike/ACE2 interaction active on entry with a high degree of standardization, easily allowing automation in a standard bio-safety environment.

## 1. Introduction

Only a small subset of the overall antibodies produced against SARS-CoV-2 are neutralizing [1]. This fraction, namely neutralizing antibodies (nAbs), is crucial because it binds to viral antigens in a manner that reduces viral infectivity, in contrast to “non-nAbs” that are not protective. A large number of individuals worldwide have acquired a post-infection or vaccine-induced seroconversion against SARS-CoV-2. A majority of infected patients develop a seroconversion but with individual variations in antibody levels [2,3], whereas mRNA vaccines always induce seroconversion [4,5]. However, age, cancer and immunosuppression are sometimes associated with lower levels of antibodies after infection or vaccination [6,7]. Importantly, seroconversion does not reflect protective immunity and a report showed that 12% of sera from infected patients containing antibodies do not possess significant levels of nAbs [8]. Other studies reported nAbs in 98% of the infected individuals after six months [9], with a persistence in 89–97% of individuals after one year [10,11,12]. After mRNA vaccination, nAbs have been reported to persist at least six months after the second dose [1]. The pandemic evolution coupled to the introduction of vaccines have imposed a selection pressure, causing the evolution of mutants [13]. These mutants possess one or more mutations in their Spike protein that prevent the neutralization by antibodies generated by the previous strain. It has been reported a reduced neutralizing activity of beta-induced antibodies to delta [12]. Additionally, sera from vaccinated patients reported a 40-fold reduction of neutralizing activity against omicron [14].

Regarding these multiple parameters influencing the humoral immunity against SARS-CoV-2, it is impossible to predict if an individual is currently protected against a defined variant. Hence, assays that can detect nAbs at both individual and community levels must be developed. An ideal test for mass testing should be rapid, cheap, reproducible, automatable and performed with minimal safety precautions. Numerous analytical methods have been developed to measure anti-SARS-CoV-2 nAbs in the serum.

As the Receptor-Binding Domain (RBD) of the viral Spike was identified to be crucial for entry, it was initially suggested that antibodies against RBD could reflect nAbs. Several immunoassays were developed to detect anti-RBD antibodies [15,16]. The surrogate virus neutralization test is a competitive ELISA, where the nAbs compete with an in vitro immobilized and quantifiable RBD/ACE2 interaction. Various kits are available and are sensitive and specific [17,18,19,20], but fail to detect nAbs at low levels [21] and suffer from a high false positive rate due to the wrong detection of non-nAbs [22]. Unfortunately for these RBD-based assays, anti-RBD are not necessarily neutralizing [23] and nAbs targeting epitopes outside the RBD are known, such as the N-terminal domain of S1 [24] and the S2 domain [25]. The plaque reduction neutralization test is the gold standard: a fixed load of a living virus is exposed to the serum prior the infection of cultured cells and counting of cytopathic effects (the formed plaques). It is a manual, labor-intensive and time-consuming procedure (72–96 h) requiring a biosafety level 3, and some strains do not produce plaques. The counting of infected cells and the time of analysis can be improved by including a reporter gene into the viral genome [26,27], but it still requires a level 3 environment. Pseudotyped viruses that are replication-incompetent express a reporter gene and use the same entry mechanism as SARS-CoV-2 have been developed [28]. They require at least 48 h of infection and can be performed under more acceptable bio-safety conditions (level 2).

In vitro cell fusion between cells expressing a viral protein and cells expressing its receptor is a method used for the quantification of virus entry in target cells. The rationale of this approach is the formation of syncytia in some virus-infected tissues, induced by the fusion of infected cells with neighboring cells. Thus, syncytia are multinucleated enlarged cells induced by the surface expression of viral proteins that interact with their receptors expressed in neighboring cells, creating a fusogenic event and allowing virus spreading without the need of endocytosis. As syncytia were described in the lungs of COVID-19 patients [29], in vitro cell fusion assays between Spike-expressing cells and ACE2-expressing cells has been described and used for the study of chemical inhibitors of the Spike/ACE2 interaction [30,31,32]. One report showed the possibility to develop a biochemical assay, based on the use of a split-beta-galactosidase, and the reduction of fusion by convalescent sera [33]. Split luciferases have also been used for measuring syncytia formation through Spike and ACE2 interaction [34,35].

With the goal to target all the epitopes of the full-length Spike with a low cost, a high rapidity of execution and a high level of standardization, we have developed a cell fusion assay emitting luminescence in a Spike/ACE2 interaction-dependent manner. As luminescence is reduced by the presence of antibodies inhibiting the Spike/ACE2 interaction, it allows a sensitive and specific quantification of inhibitory antibodies in serum. This new assay is rapidly executed (24 h), reproducible, cheap, automatable and easily standardized for large-scale analyses.

## 2. Material and Methods

### 2.1. Cells and Reagents

Hela cells were cultured in DMEM medium with 4.5 g/L of glucose (Gibco), supplemented with 1% of penicillin and streptomycin, non-essential amino-acids and 1 mM sodium pyruvate and 10% of fetal bovine serum (Thermofisher, Waltham, MA, USA). The used luciferase substrate for the detection of luminescence was The NanoGlo live assay, from Promega, Madison, WI, USA.

### 2.2. Blood Samples

Individuals were recruited among the STRAT-CoV and GEROCOVID cohorts, including patients hospitalized for COVID-19 at the Geneva University Hospitals, Switzerland. SARS-CoV-2 infection was diagnosed with a positive Reverse Transcriptase—Polymerase Chain Reaction against SARS-CoV-2 in naso-pharyngeal swabs. Blood samples were collected at admission and about two weeks later. This study also included volunteers sampled at different time points depending of the disease and vaccinations with mRNA vaccines. Patients with psychiatric disorders or for whom consent could not be obtained were excluded. Ethics approval was granted by the Cantonal Ethics Research Committee of Geneva, Switzerland (GEROCOVID: no. 2020-01248, Dr V. Prendki and Dr A. Malézieux-Picard, April 2020-May 2021; STRAT-CoV: no. 2020-01070, Dr S. Baggio, Dr N. Vernaz, February 2020–February 2022). A serology was also performed with some COVID-19 positive patients, by using the Roche (Roche Diagnostics International Ltd., Rotkreuz, Switzerland) anti-SARS-CoV-2 IgG quantitative ECLIA kit. Results were automatically reported as the analyte concentration of each sample in U/mL, with <0.80 U/mL interpreted as negative for SARS-CoV-2 anti-S Ig antibodies and ≥0.80 U/mL interpreted as positive for SARS-CoV-2 anti-S Ig antibodies.

### 2.3. Molecular Biology

The ACE2-expressing Hela cells (Hela-ACE2) were established from a cDNA ORF purchased from GenSript. The pCG1_SCoV2-S plasmid encoding the original Wuhan Spike (S_wuhan_) (provided by Prof. Dr. Stefan Pöhlmann, University Göttingen, Göttingen, Germany) was used to generate the Spike-expressing Hela cells (Hela-S). ACE2 and Spike cDNA ORFs were cloned into pCDH-CMV-MCS-EF1α-Puro using a standard cloning method. Four different variants of Spike were also generated with the indicated mutations: S_D614G_ (D614G), S_delta_ (E156-F157del, R158G, L452R, T478K, D614G, P681R, D950N), S_lambda_ (G75V, T67I, R246-G252del, D253N, L452Q, F490S, D614G, T859N) and S_omicron_ (A67V, H69-V70del, T95I, G142-V143-Y144del, Y145D, N211del, L212I, G339D, S371L, S373P, S375F, K417N, N440K, G446S, S477N, T478K, E484A, Q493R, G496S, Q498R, N501Y, Y505H, T547K, D614G, H655Y, N679K, P681H, N764K, D796Y, N856K, Q954H, N969K, L981F).

The plasmids SmBiT-PRKACA and LgBiT-PRKAR2A containing, respectively, the small BiT and the large BiT part of the luciferase were purchased from Promega (NanoBiT^®^ PPI MCS Starter System CAT.# N2014). SmBiT-PRKACA and LgBiT-PRKAR2A cDNA ORFs were cloned into pCDH-CMV-MCS-EF1α-RFP and pCDH-CMV-MCS-EF1α-CopGFP, respectively, using standard cloning methods.

For recombinant-lentivirus production, lentivector plasmids were transfected in HEK 293T cells using the calcium phosphate method. Hela-S and Hela-ACE2 were established from lentiviral cotransduction resulting in two cell lines expressing, respectively, Spike/GFP/LgBiT and ACE2/RFP/SmBiT.

### 2.4. Cell Fusion Assay

A mixture of 10,000 clonal Hela-S and 10,000 clonal Hela-ACE2 were cocultured in 96-well flat clear bottom white polystyrene microplates (Corning) in 100 µL of culture medium in the presence of patient’s serum or control sera at various dilutions (1/8, 1/32, 1/128 and 1/512). After 24 h of culture at 37 °C under 5% of CO_2_, the medium was removed and cells were rinsed once with HBSS buffer with Calcium and magnesium (Thermofisher) prior to the addition of 100 µL of the same HBSS buffer. Twenty-five microliters of the substrate diluted at 1/20 in buffer (NanoGlo live assay, Promega, see the manufacturer’s instructions) was added extemporaneously. The microplate was shook for 1 min at 500 rpm, then centrifugated 1 min at 300× *g* prior to the addition of an adhesive white opaque film at the bottom of the plate and luminescence was measured with a spectra-L-luminometer (Molecular devices). Experiments were performed in triplicates. To prevent variability between operators, assays and reagents, the same internal control made of pooled sera from SARS-CoV-2—negative patients were systematically included. The inhibition was normalized for each serum dilution with this internal control and calculated as the ratio between the luminescence from patient’s serum and the luminescence from the control serum.

### 2.5. Competitive ELISA for the RBD/ACE2 Interaction

Sera were diluted in assay buffer at the same dilutions as for cell fusion assay (1/8, 1/32, 1/128 and 1/512) and analyzed by using the SARS-CoV-2 Neutralizing Ab ELISA Kit (ThermoFisher) according to the manufacturer’s instructions.

### 2.6. Pseudotyped-Based Assay for the Detection of nAbs

For Spike-pseudotyped lentivirus production, plasmids were transfected in 293T cells using the calcium phosphate method. Briefly, 4.5 × 10^6^ cells were plated in a 10-cm dish and transfected 16 h later with 15 μg of pCDH-CMV-Gluc-EF1α- GFP, 10 μg of packaging plasmid [psPAX2, gift from Didier Trono (Addgene plasmid 12260)] and 5 μg plasmid encoding Spike D614 DeltaCter under control of the cytomegalovirus promoter (pCG1_SCoV2-S DeltaCter D614G). The medium was changed 8 h post-transfection. After 48 h, the viral supernatants were collected and filtered using 45 µm PVDF filters and stored at −80 °C. Hela were seeded into 96-well plates at a density of 3000 cells per well. Then, 16 h later, 25 μL of serial dilutions of patient’s sera or control sera were incubated with 75 μL of media containing Spike pseudovirions for 30 min. After the incubation, the mix was added to the cells. After 24 h incubation, cells were washed with PBS and the media was refreshed. Three days later, supernatants were collected to measure luciferase activity.

## 3. Results

Coculture of Hela cells expressing the SARS-CoV-2 Spike and Hela expressing ACE2 induces cell fusion.

Hela cells were stably transduced with the original Wuhan Spike sequence (S_wuhan_) or some selected variants: the European variant with the D614G mutation (S_D614G_), lambda variant (S_lambda_), delta variant (S_delta_) and omicron variant (S_omicron_). Other Hela cells were stably transduced with the ACE2 receptor, also under the control of a ubiquitous promoter. To allow quantification of fusion, a dual split reporter system has been also introduced in Hela-S and Hela-ACE2 lines. The Hela-S lines were co-transduced with a construct expressing a large part of the luciferase, namely, the large BiT luciferase (LgBiT) and the Green Fluorescent protein (GFP), whereas Hela-ACE2 lines were co-transduced with a construct co-expressing the complementary resting small part of luciferase, namely small BiT luciferase (SmBiT) [1] and the Red Fluorescent Protein (RFP). Upon fusion, the expressed LgBiT and SmBiT recombine and produce a functional luciferase emitting quantifiable luminescence in the presence of a cell-permeable substrate (Figure 1A). Cultured alone in a 96-well plate, clonal Hela-S or clonal Hela-ACE2 developed towards a monolayer of single cells (Figure 1B,C). In contrast, the coculture rapidly (24 h later) induced cells clusters, suggesting fusogenic events and formation of syncytium-like structures (Figure 1D). These clusters were double-fluorescent for GFP and RFP, whereas Hela-S alone were GFP+/RFP- and Hela-ACE2 were GFP-/RFP+ (Figure 1E), confirming cell fusions between the two lines. Expectedly, the replacement of Hela-S_wuhan_ by Hela-S_D614G_, Hela-S_delta_ and Hela-S_lambda_ increased fusion because all of these variants have more affinity for ACE2. Indeed, the in vitro fusogenic events were strong enough to create rapidly bigger cell clusters having the tendency to detach from the bottom of the plate (Figure 1F). In contrast, Hela-S_omicron_ induced clusters similarly to Hela-S_wuhan_.

With the goal to develop an assay measuring antibodies inhibiting the Spike/ACE2-dependent cell fusion, serum must be added to the coculture prior to washing and addition of a cell-permeable substrate for luciferase. Washing is mandatory because human serum is known to inhibit the reaction between the luciferase and its substrate. Because detachment of cell clusters, observed with some variants of Spike, must be avoided at the washing step (to ensure reproducibility and use in a large-scale setting), we decided to develop and evaluate an assay using the Hela-S_wuhan_.

Cells cultured alone in a 96-well plate (Hela-S-LgBiT alone or Hela-ACE2-SmBiT alone) did not generate luminescence, in contrast to the co-culture (Figure 2A). The signal increased with the cell concentration, confirming a dose-dependent response (not shown). Co-culture using a Hela cell expressing LgBiT but not Spike (Hela-LgBiT) abolished the signal, confirming that the emission of the fluorescence was dependent on the Spike/ACE2 interaction (Figure 2A).

### 3.1. Coculture of Hela-S and Hela-ACE2 in the Presence of Various Dilutions of Seronegative Human Serum Induces Emission of a Luminescent Signal

Because the goal of the assay is to evaluate, in human serum, the presence of antibodies inhibiting Spike/ACE2-dependent fusion, the impact of human serum on the emission of luminescence was first evaluated. Indeed, factors present in the human serum could interfere with the cellular fusion, independently of immunoglobulins. Several serum dilutions ranging between 1/8 and 1/512 were tested by using samples from 29 seronegative patients without any history of COVID-19 or vaccination (GEROCOVID cohort). To prevent variability between operators, assays and reagents, an internal control made of a pooled seronegative serum was systematically included. Each luminescent signal was then normalized by the internal control (in %). Luminescence emission was proportionally reduced by the addition of increased concentrations of Alpha1 Anti-Trypsin (AAT), an inhibitor of TMPRSS2 protease [36] (Figure 2B) indicating that the fusion was dependent on the processing of Spike and not solely its physical interaction with ACE2. The results from 29 seronegative individuals and for each serum dilution are shown. Inhibition of fusion was expressed in percentage of the internal control serum (Figure 2C). One patient was considered to be an outlier (probably an unknown seropositivity). The 1/8 dilution showed the highest variability (standard deviation) between patients, suggesting that an excess of serum could impact the reproducibility of the test (Figure 2C). The calculation of two standard deviations from the average was used to define a cut-off value for the detection of a statistically significant reduction of fusion. The value of the cut-off was impacted by the serum dilution at 1/8 due to the increased variability, in contrast with higher dilutions. Table 1 shows the numerical values of the percentage of inhibition and two standard deviations from the average for each serum dilution.

### 3.2. Impact of Seropositive Sera from COVID-19 Patients on Hela-S and Hela-ACE2 Fusion

In total, 54 sera from COVID-19 patients (D614G European variant) were collected at the Geneva University Hospitals, Switzerland, two weeks after the first symptoms, thanks to the STRAT-CoV cohort (Dr V. Prendki and Dr A. Malézieux). A proportion of these sera showed neutralization higher than the cut-off of two standard deviations defined previously (Figure 3). An effect of the serum dilution was observed: the more the serum was concentrated in the reaction, the more intense the neutralization was (on average), with a maximum at the 1/8 dilution (Figure 3). Because the cut-off was higher at this dilution, the 1/32 dilution was probably the best one for the interpretation of the data. Thus, compared to seronegative sera, sera from COVID-19 patients were able to reduce the Spike/ACE2-dependent fusion, suggesting the presence of inhibitory antibodies. We also collected sera from 23 patients (GEROCOVID cohort, Dr V. Prendki and Dr A. Malézieux) at two different time points: at the moment of first symptoms of a primo-infection (T1) and 2 weeks later (T2), considering the absence of humoral immunity at T1. These 23 patients were not vaccinated. For a majority of these patients, there was a clear increase in fusion inhibition at T2 compared to T1 (Figure 4), confirming that the inhibition was acquired 2 weeks after the first symptoms and not constitutive of the nature of the serum, and therefore, again suggesting fusion-inhibitory antibodies.

### 3.3. Comparison of the Fusion-Based Assay with an Assay Measuring the Inhibition of RBD/ACE2 Interaction, Serology and Pseudotyped-Based Assay

A comparison between fusion-targeting Abs and Abs targeting the RBD was next performed because competitive methods exploiting the physical interaction between ACE2 and RBD are the more standardized and currently used assays. Forty-seven serologically negative or positive sera were analyzed either for the presence of fusion-targeting Abs and RBD-targeting Abs, and correlation graphs were established for each serum dilution. The competitive ELISA used for RBD-targeting Abs was the SARS-CoV-2 Neutralizing Ab ELISA Kit (ThermoFisher). In total, 47 serologically negative sera were tested and completed by 33 serologically positive sera from infected patients. The cut-off value of positivity for nAbs with the competitive ELISA was 20% of inhibition (according to the manufacturer), whereas with the fusion assay, as established previously, it was dependent on the serum dilution. At the 1/8 dilution (showing the highest variability and the highest cut-off), both methods detected neutralization for the high percentages of inhibition, but with a low correlation coefficient (R^2^ = 0.19) (Figure 5). For this low dilution of serum, the 14 serologically negative samples were all detected by the competitive ELISA and only 2/14 by the fusion, confirming an important false-positivity rate of the competitive ELISA when serum is more concentrated. At the 1/32 dilution, there was again a low correlation between the two methods (R^2^ = 0.25). In total, 4/14 serologically negative sera were detected as positive only by the competitive ELISA and 3/14 by the fusion assay. Two serologically positive samples were detected by the competitive ELISA, but not the fusion assay. At 1/128 dilution, the correlation remained low (R^2^ = 0.27), with eight serologically positive sera only detected by the RBD/ACE2 competitive ELISA. At this dilution, three positive sera were still detected by the fusion, but lacked their detection by the competitive ELISA. At the higher serum dilution (1/512), the sensitivity of the two tests decreased enough to induce a lack of detection of many samples. We also performed comparisons between the fusion assay and the serology, as well as the RBD/ACE2 competitive ELISA and the serology. The serology was performed by using bioluminescence (anti-SARS-CoV-2 IgG quantitative ECLIA kit, Roche Diagnostics). Table 2 summarizes de correlation coefficients, for each serum dilution, observed with the three comparisons: fusion vs RBD/ACE2 competitive ELISA, fusion vs serology and RBD/ACE2 competitive ELISA vs serology. The correlation between the different tests was relatively poor, with R^2^ values between 0.2 and 0.4 demonstrating that different types of antibodies are detected with the different tests. There was a trend for a better correlation between serology and RBD/ACE2-competitive ELISA, as compared to the other correlations; however, no statistically significant differences were observed (William’s t test). The correlation graphs between fusion and serology are shown in Appendix A. The best observed correlations were between the RBD/ACE2 and the serology, whereas the fusion assay was lowly correlated with serology and RBD/ACE2 competitive ELISA. Notably, serology and RBD/ACE2 detected antibodies and neutralization without any impact on fusion.

The fusion-based assay was also compared with an infection assay using pseudotyped lentivectors expressing Spike. Height sera were analyzed by fusion versus pseudotyped. Pseudotyped lentivectors expressed the same Spike than Hela cells used in fusion, and also a complete luciferase under the control of a ubiquitous promoter. In contrast to correlations between fusion/RBD-ACE2 and between fusion/serology that were very weak, the correlation between fusion/pseudotyped was higher. Figure 6 shows the correlation graphs and Table 2 reports the coefficient correlations for three serum dilutions. Thus, the fusion-based assay is more correlated with an infection assay exploiting entry than a competitive ELISA or a conventional serology. The sensitivity of the fusion assay to detect inhibition was observed to be weakly lower than the pseudotyped assay. Together, these observations reinforce the fact that the fusion-based assay more exploit a functional entry process rather than a biochemical inhibition of Spike with its receptor. Entry is a more relevant target for the detection of antibodies active on infection.

### 3.4. Case-Reports of the Kinetic of Fusion Inhibitions in Vaccinated Individuals

Because of the large-scale introduction of vaccination worldwide, it was important to evaluate the inhibition of fusion after vaccination. Figure 7 shows several case-reports. A first individual (ID#1) was primo-infected by the beta variant and did not show any neutralization below the cut-off three weeks after the first symptoms. A mRNA vaccine following primo-infection induced a stable inhibition (below the cut-off) for at least 5 months, maintained by a second mRNA vaccine later. This confirms that fusion-inhibitory antibodies can be detected and followed after mRNA vaccination. Some individuals (ID#2 to ID#5) vaccinated by various series of mRNA vaccines without any previous primo-infection showed stable fusion-inhibitory antibodies after the first vaccination for at least 6 months. On the other hand, other individuals also vaccinated by series of mRNA vaccines (ID#6 to ID#8) only showed inhibition after the second dose of vaccine. It is noteworthy that one individual (ID#9) lost fusion-inhibitory antibodies 8 months after the second dose. Appendix A shows the details of the vaccinal scheme of each individual.

Together, these observations indicate that the assay can be used for the determination of fusion-inhibitory antibodies following vaccination. As expected, there is a clear variability in the vaccinal response between patients, reinforcing the need of new assays evaluating the inhibition of the Spike/ACE2 interaction.

## 4. Discussion

Cell-to-cell fusion occurs in vivo because syncytia, which are large multinucleated pneumocytes, are seen in the lungs of COVID-19 patients [37]. Mechanistically, cell-to-cell fusion is similar to virus-cell fusion, and thus, represents an excellent model of viral entry, reinforcing the interest of its use for the detection of inhibitory antibodies. The viral Spike is considered to be the fusogenic molecule. Viral entry begins when Spike interacts with ACE2. Then, Spike is cleaved by TMPRSS2 at the S1/S2 junction site to allow fusion in a zone of the plasma membrane to create an endocytic vesicle engulfing the virus. Viral RNA is transferred to the cytoplasm and translated, producing a Spike protein translocated into the endoplasmic reticulum and transported throughout the Golgi and the membrane. The membrane Spike associates with a ACE2 receptor on neighboring cells via its RBD domain, and is also again processed by proteases at the S1/S2 sites. S1 domain is released, which allows the fusion process between the two cells through the formation of a common pore that will expand [37]. Thus, developing an assay measuring inhibitors of fusion will add a more functional evaluation of the impact on viral entry mechanisms rather than only a Spike/ACE2 or RBD/ACE2 physical interaction. Accordingly, to this point, we observed that the cell fusion assay performed in the presence of inhibitors of TMPRSS2 reduced fusion, confirming the implication of the molecular machinery of entry. Hela cells do not express TMPRSS2 (not shown); thus, other extracellular protease-mediated pathways might be important with these cells. Metalloprotease-mediated cell fusion has recently been reported [38,39] and should be explored in further studies by using specific protease inhibitors.

If the Pseudotyped virus entry assay also has the advantage of modeling the Spike-dependent virus/entry mechanism, the fusion assay has the strong advantage of reduced cost, simplicity and not requiring a bio-safety level 2 environment.

The degree of syncytia formation correlates with the affinity of Spike with ACE2 [40,41], and we observed that changing the original Wuhan Spike in the fusion assay by other variants strongly influenced fusogenicity, with a strong increase seen with the European (D614G), delta and lambda variants. The European variant contains a mutation (D614G) close to the S1/S2 cleavage site, conferring to Spike a more competent binding for ACE2 [42,43]. Pseudovirus assays performed by our group (not shown) and others [44,45] have shown that the D614G mutation increases the efficiency of viral entry. Moreover, the D614G mutant produces more syncytia in vitro than the Wuhan strain [41,46]. In accordance with our observations, the delta variant is more fusogenic than the D614G variant [41,47,48]. The impact of the lambda and omicron mutations on fusion are not described precisely and our data show an impact of lambda, but not omicron.

Because a too strong intensity of fusion induces big clusters detaching from the plate, specific technical adaptations need to be achieved to assess each variant in the assay. We used the Wuhan sequence in this study as a proof of concept, expectedly exploitable for patients infected by the delta variant and individuals vaccinated with the most current mRNA vaccines. The possibility to adapt easily any variants will represent a strong advantage. The interest in using various Spike mutants in the assay is useful not only for the evaluation of the protective immunity of some populations exposed to defined variants or new mRNA vaccines, but also to understand the immunological cross reactivities between the variants and vaccines.

Another advantage of this fusion assay consists in its rapidity of execution (24 h) in multiwell plates, compared to the plaque reduction neutralization test and pseudotyped viruses, and in a standard bio-safety environment. It then provides a strong level of reproducibility and standardization adapted for automation and large-scale studies.

There was a clear impact of human serum by itself on fusion, independently of the presence of inhibitory antibodies. For some sera, serum dilution is associated with a trend for a slightly increased fusion. As this trend is not statistically significant, we have not investigated possible underlying mechanisms.

Regarding the comparison of the fusion-based method and competitive ELISA exploiting the RBD/ACE2 interaction, some conclusions were established. For low serum dilution (1/8), the competitive ELISA was not specific because all of the seronegative samples were detected. In contrast, the fusion-based assay was more specific in these conditions; however, it shows some false-positive results (2/14). Thus, seric factor in abundance probably interfere with the RBD/ACE2 interaction to generate false-positive results. At higher serum dilutions (1/32 and 1/128), there was a weak correlation (R^2^~0.2) between the two methods with a comparable sensitivity. A correlation with a R^2^ of 0.2 means that only for 20% of the studied samples there was an association between the RBD/ACE2 ELISA and the fusion-targeting Abs. In general, a clinically relevant association has a R2 value of >0.8. Thus, our results strongly suggest that the functional characteristics of antibodies detected by fusion are for large parts different from the functional characteristics of antibodies detected by the RBD/ACE2 ELISA. We hypothesize that the fusion-targeting Abs are most relevant to viral entry; however, larger-scale clinical studies will be required to specifically address this question.

How do we interpret the data to produce numerical values evaluating the neutralization? First, there is the question of the cut-off value, which will depend on variants and must be determined in each assay. Two standard deviations below the average of at least 30 seronegative sera appear to be a statistically correct cut-off to define the presence of an inhibitor. We observed a clear effect of serum dilution on the obtained value, which was also dependent on each individual. Thus, we would not preconize to select an arbitrary dilution of serum and just render a result in percentage of luminescence compared to the internal control serum. It would represent a risk to overestimate or underestimate the inhibition, because outside of the optimal window. Thus, we more preconize the systematic use of several dilutions and the calculation of the dilution that induce a reduction of 50% of the control signal (namely the half inhibitory concentration or IC50).

## 5. Conclusions

We describe in this study a new dual split luciferase-based fusion assay to measure the presence of fusion-inhibitory antibodies in the serum of individuals. It will allow large-scale screening of immunity in populations post disease or vaccination or treatment. This assay is cheap, well standardized for automation and does not require a specific bio-safety environment. Because it uses a full-length Spike protein and includes molecular events mimicking viral entry, this assay can be performed with variants of viral fusion proteins that also lead to cell-to-cell fusion. Detection of fusion-targeting Abs is not only relevant to viral entry, but also the post-entry pathogenesis. Indeed, cell-to-cell fusion and generation of multinucleated cells is part of the COVID pathogenesis and also observed in other types of viral infections (e.g., HIV).

## Figures and Tables

**Figure 1 viruses-14-02118-f001:**
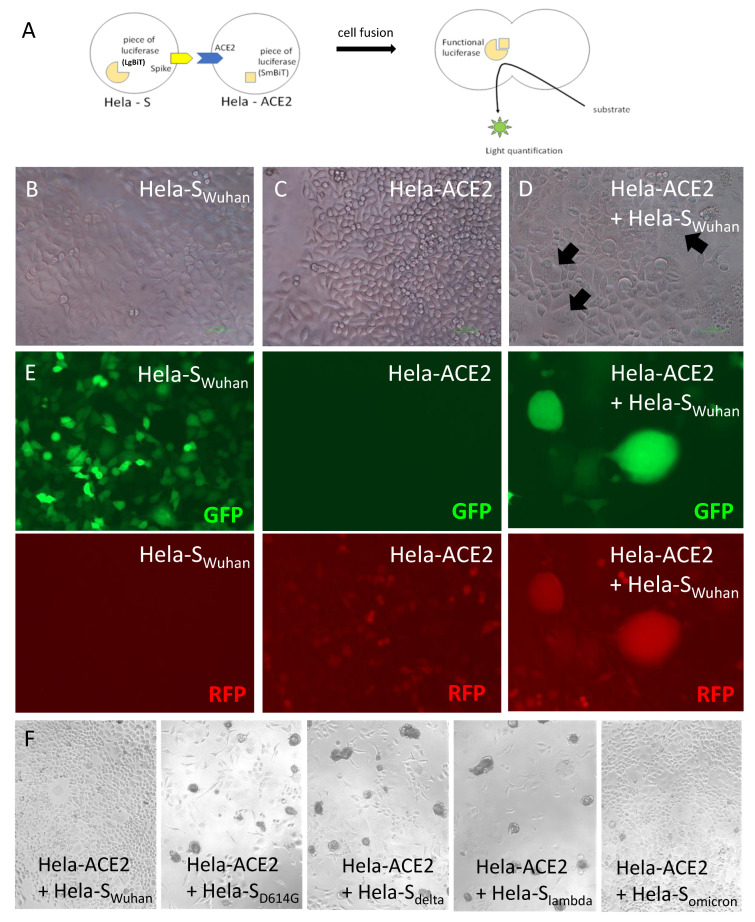
Coculture of Hela-S and Hela ACE2 induces cell fusion. (**A**) Principle of the dual split fusion assay between a Hela-S and a Hela-ACE2. B-F Cells were cultured in 96-well plate for 24 h. (**B**–**D**) Microscopic observation of Hela-S (**B**) or Hela-ACE2 (**C**) alone versus co-culture (**D**). The black arrows show cell syncitia indicative of fusion. Magnification ×40. (**E**) Observation of the fluorescence emitted by Hela-S or Hela-ACE2 alone versus co-culture. Hela-S were GFP+ and Hela-ACE2 RFP+. Magnification ×40. (**F**) Hela cells expressing various variants of Spike were cocultured with Hela-ACE2, prior to a microscopic observation of cells. Magnification ×20.

**Figure 2 viruses-14-02118-f002:**
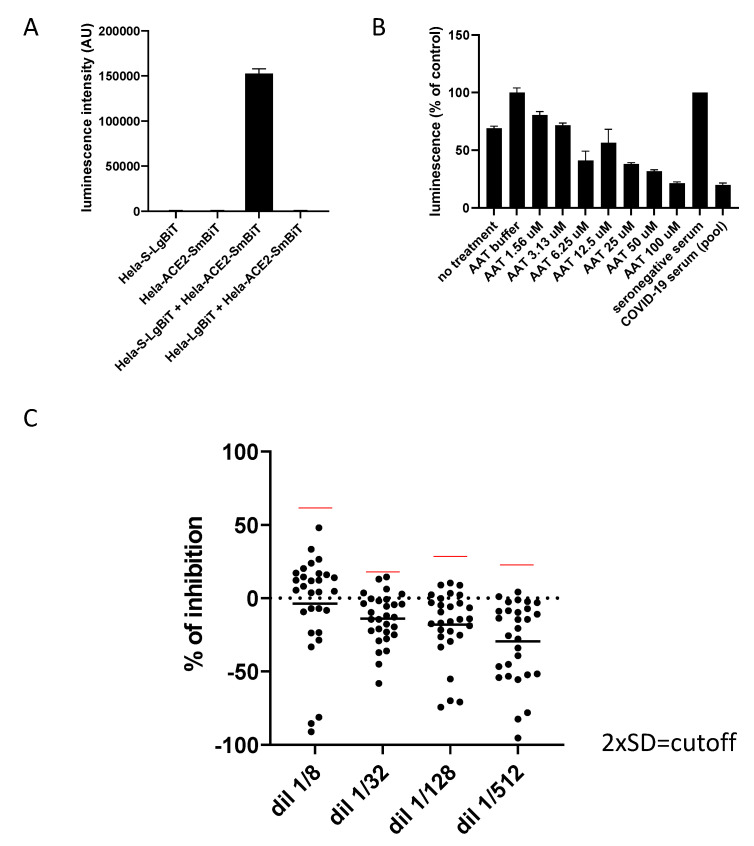
Impact of human seronegative serum on fusion. (**A**) Various combinations of Hela transduced or not with S, ACE2, LgBiT or SmBiT were cocultured 24 h in a 96-well plate, prior to the addition of the substrate for luciferase and reading on a luminometer. (**B**) Influence of AAT on cell fusion. Each luminescent signal is normalized with the value in the presence of an internal control, made of another pool of seronegative sera (expressed in %). (**C**) Coculture of Hela-S and Hela-ACE2 in the presence of various dilutions of sera (1/8 to 1/512) from seronegative patients without any COVID-19 history. Each luminescent signal is normalized with the value in the presence of an internal control, made of another pool of seronegative sera. Results are expressed as the percentage of inhibition compared to the internal control serum.

**Figure 3 viruses-14-02118-f003:**
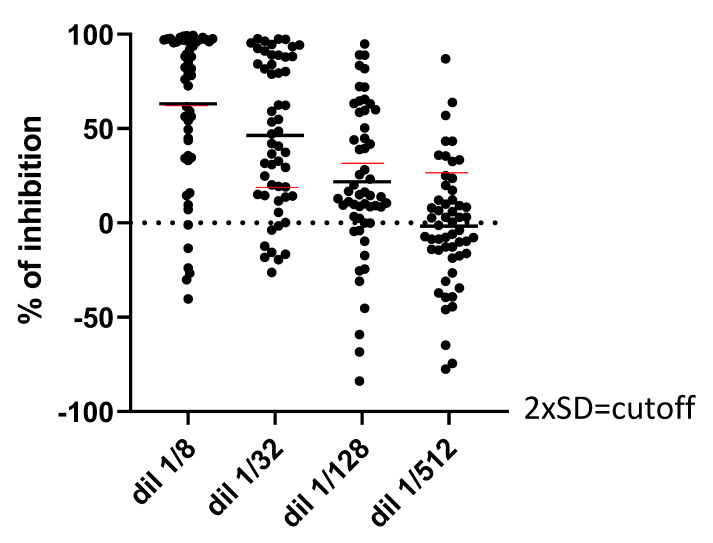
Impact of serum from COVID-19 patients on fusion. Coculture of Hela-S and Hela-ACE2 in 96-well plate for 24 h in the presence of various dilutions of sera (1/8 to 1/512) from COVID-19 patients two weeks after the first symptoms. Each luminescent signal is normalized with the value in the presence of an internal control, made of a pool of seronegative sera. Results are expressed as the percentage of inhibition compared to the internal control serum.

**Figure 4 viruses-14-02118-f004:**
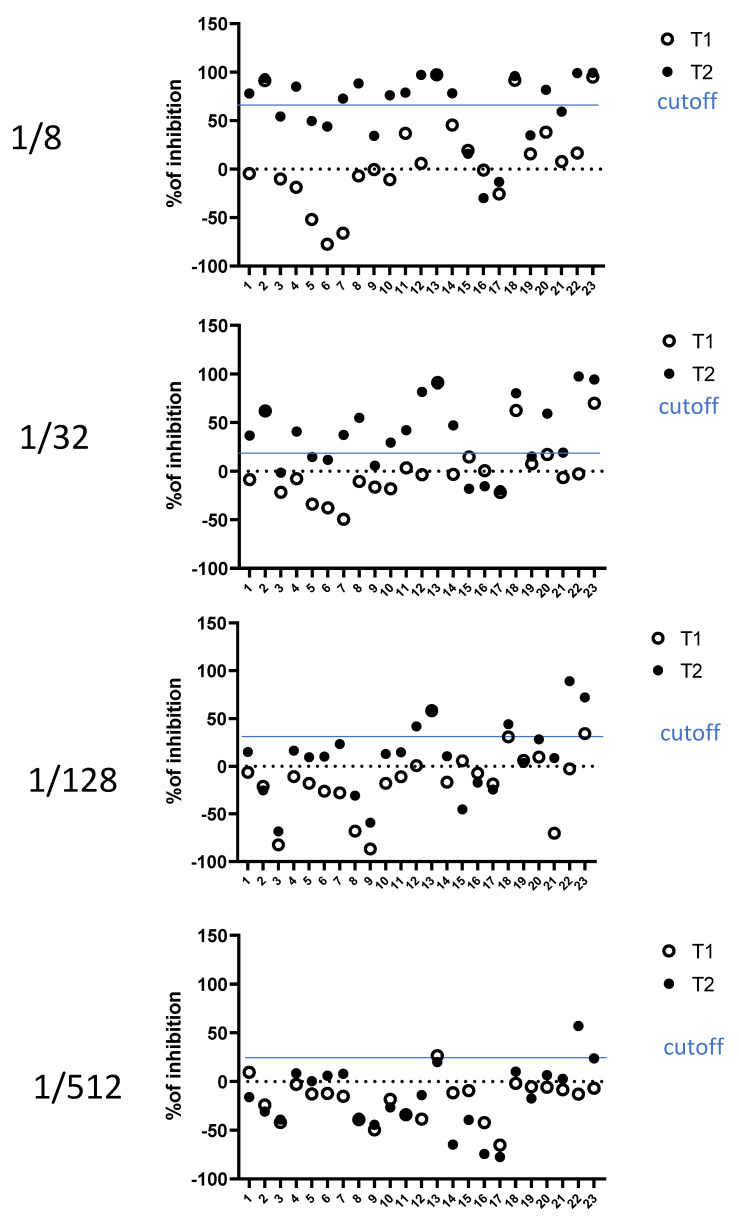
Impact of sera from COVID-19 patients on fusion at the moment of first symptoms and 2 weeks later. Coculture of Hela-S and Hela-ACE2 in the presence of various dilutions of sera (1/8 to 1/512) from COVID-19 patients at the moment of first symptoms (T1) or two weeks after (T2). Each luminescent signal is normalized with the value in the presence of an internal control, made of a pool of seronegative sera. Results are expressed as the percentage of inhibition compared to the internal control serum.

**Figure 5 viruses-14-02118-f005:**
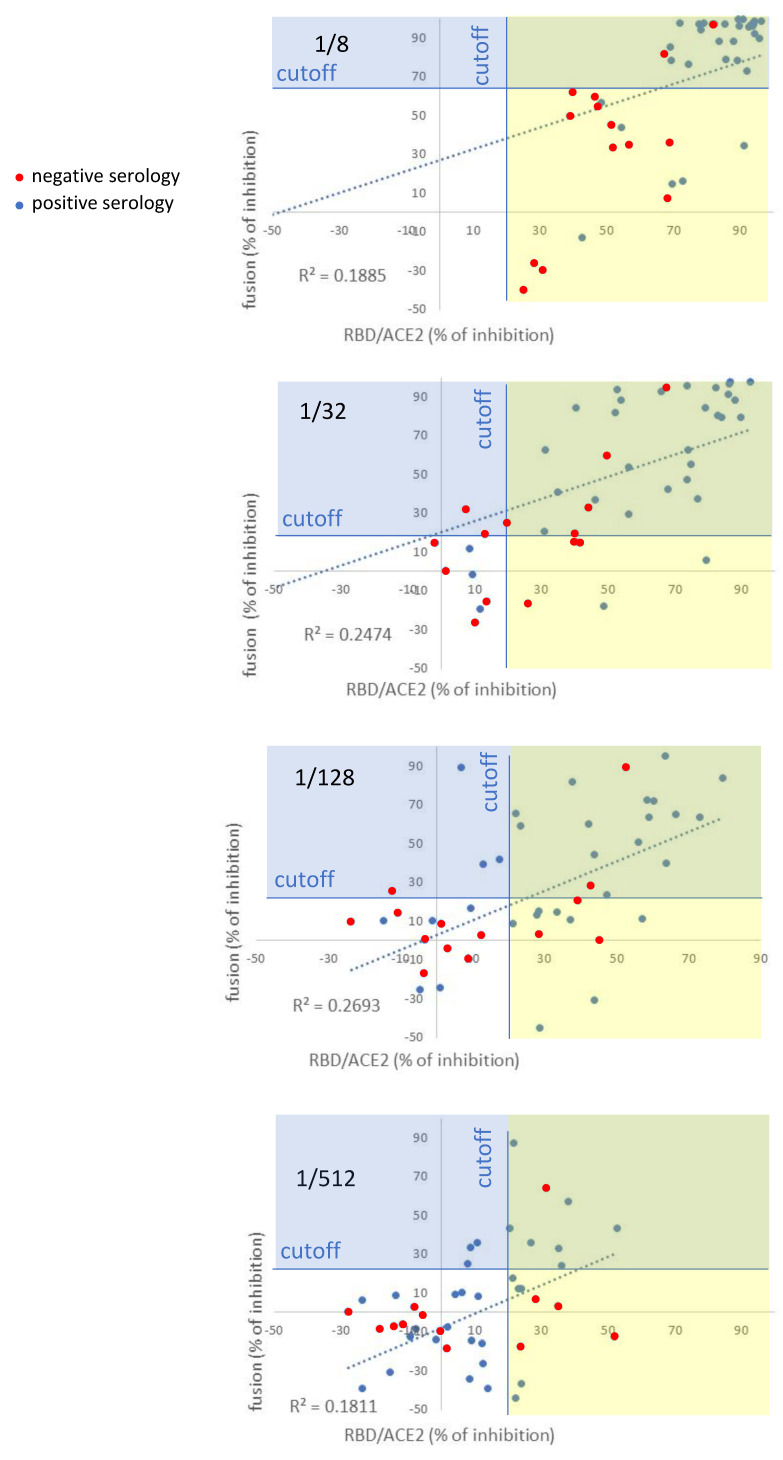
Comparison of the fusion-based assay with a competitive ELISA measuring the RBD/ACE2 interaction. In total, 47 sera were either analyzed, for each dilution, with the fusion-based assay or a competitive ELISA measuring the RBD/ACE2 interaction. The cut-off value for the ELISA is 20% of inhibition (as indicated by the manufacturer).

**Figure 6 viruses-14-02118-f006:**
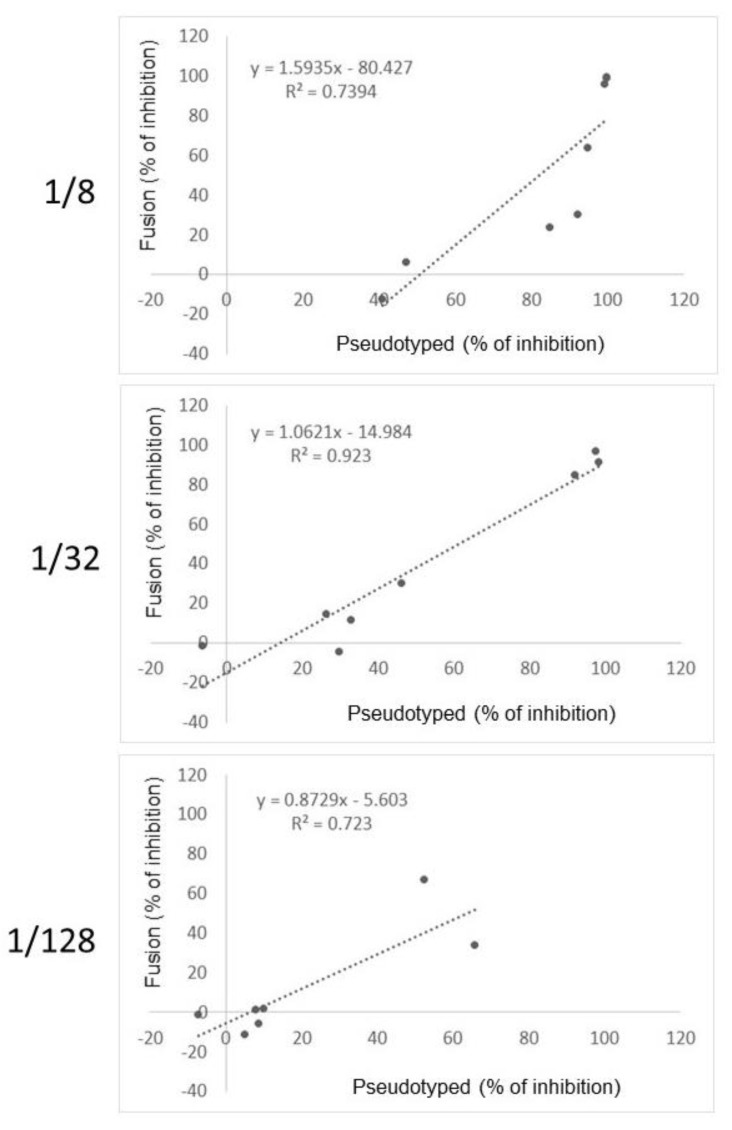
Comparison of the fusion-based assay with a pseudotyped-based entry assay. Height sera were either analyzed, for each indicated dilution, with the fusion-based assay or an assay using a Spike/luciferase-expressing lentivector.

**Figure 7 viruses-14-02118-f007:**
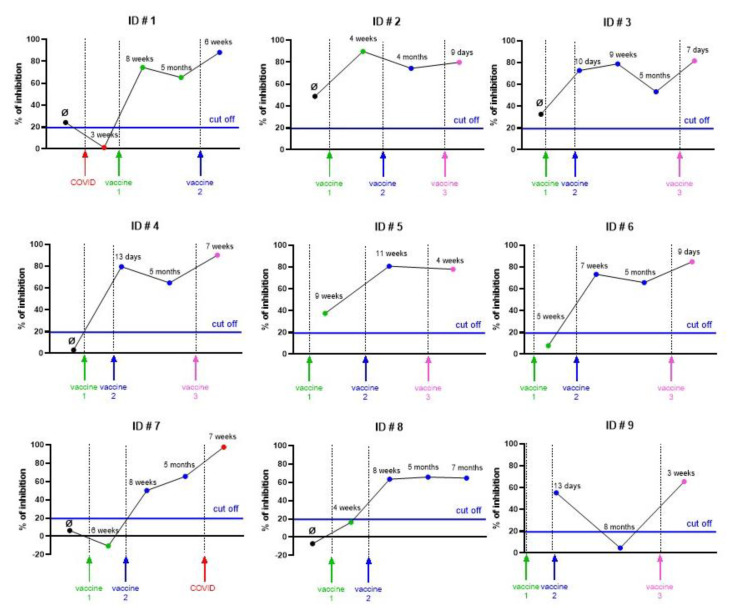
Case-reports of fusion inhibition after mRNA vaccination. Sera (1/32 dilution) from nine volunteers (**ID#1** to **ID#9**) with various vaccinal schemes were analyzed for their impact on the fusion inhibition. Ø = serum before any vaccination/infection. The time of vaccination or infection is indicated by vertical arrows with a color code (red for COVID infection, green for first vaccination, blue for second vaccination and pink for third vaccination). The dots correspond to blood sampling and the time laps refer to the delay between blood sampling and the previous vaccination/infection. Each luminescent signal is normalized with the value in the presence of an internal control, made of a pool of seronegative sera. Results are expressed as the percentage of inhibition compared to the internal control serum.

**Table 1 viruses-14-02118-t001:** Average and standard deviation between 29 non-immune sera tested for their impact on the fusion (expressed in percentage of inhibition). No statistically significant impact of non-immune sera on fusion was observed.

Serum Dilution	Average	2 Standard Deviations from the Average
1/8	−3.7	62.6%
1/32	−14.1	19.4%
1/128	−18.1	27.8%
1/512	−29.5	24.1%

**Table 2 viruses-14-02118-t002:** Coefficient of correlation (R^2^) obtained from different comparisons between the RBD/ACE2 competitive ELISA, the fusion assay and a conventional serology using bio-chemiluminescence. In total, 47 sera were tested by the three methods and correlations coefficients were calculated. The 1/512 dilution was removed because of a lack of sensitivity of detection with this dilution.

Serum Dilution	Fusion vs. RBD/ACE2	Serology vs. RBD/ACE2	Fusion vs. Serology	Fusion vs. Pseudotyped
1/8	0.19	0.36	0.20	0.74
1/32	0.25	0.40	0.28	0.92
1/128	0.27	0.38	0.20	0.72

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
