# Peer review of "A Cellular Assay for Spike/ACE2 Fusion: Quantification of Fusion-Inhibitory Antibodies after COVID-19 and Vaccination"

_viruses, 2022, doi:10.3390/v14102118_

Round 1

Reviewer 1 Report (Previous Reviewer 2)

The authors adequately responded to the comments. I have no further comments and congratulate the authors.

Minor comments:

1. At 1/128 dilution I counted 8 serologically positive sera only detected by the RBD/ACE2 (instead of 7).

2. In the same paragraph a spelling mistake is in the 3rd row from the bottom (RBD instead of RB2).

Author Response

Reviewer 2 Report (New Reviewer)

The authors established an assay system to quantify membrane fusion of SARS-CoV-2 using a cell fusion assay with HeLa cells and a split reporter protein. The assay system to quantify membrane fusion of SARS-CoV-2 with a split reporter protein is not new, as many studies have been reported.

However, the authors performed precise studies, including evaluation of AAT concentrations and determination of cutoff values using negative control samples.

The authors need to report more details on this assay system since no novel serological results were obtained from the present paper.

Since cell fusion and viral infection share many common mechanisms but also differ in some parts, the properties of the authors' assay system should be clarified in more detail. Further discussion is needed, such as the low correlation between ELISA and fusion assay.

Major point

1.    The authors have evaluated Hela-ACE and Hela-Spike cell fusions, but generally Hela cells do not express TMPRSS2. Given that endosomal CTPL-mediated fusion does not contribute to cell fusion, other extracellular protease-mediated pathways may be important in this cell. Metalloprotease-mediated cell fusion has recently been reported. The membrane fusion pathway in Hela cells should be clarified by using several protease inhibitors.
mBio, 2022 Aug 30;13(4):e0051922.
J Virol. 2021 Apr 12;95(9):e00002-21.

2.    Low correlations were observed between the cell fusion assay and the ELISA, but the most important aspect is the correlation of each assay with the ability to inhibit viral infection. In general, cell fusion is less sensitive to inhibitors than viral infection, so more antibodies are required for cell fusion assay. On the other hand, ELISA often uses the RBD, which is a part of spike protein, and cannot accurately evaluate the activity of antibodies that bind to the spike N-terminal domain outside the RBD. The authors should discuss the problems with each assay system and, if possible, perform infection assays on several samples to evaluate actual activity.

Round 2

Reviewer 2 Report (New Reviewer)

We thank the authors for their appropriate corrections.

I agree with the authors that they add a discussion on proteases other than TMPRSS2 without performing new experiments for this paper. However, the identification of proteases for entry in HeLa cells should be analyzed in the future, since the effect of neutralizing antibodies and inhibitory proteins in serum samples may depend on the protease used for entry. 

In Fig.6 and Table.2, important information about the advantages of cell fusion assays has been presented and will be of great benefit to the reader.

Please provide detailed protocols for the preparation of pseudoviruses, including information on the promoter of the Spike expression plasmid and time to collect the viral solution.

Author Response

Dear Editor,

Thank you to have reviewed the enclosed revised manuscript “A cellular assay for spike/ACE2 fusion: quantification of fusion-inhibitory antibodies after COVID-19 and vaccination” we would like to resubmit to Viruses, special issue “Novel Diagnostic Technologies for SARS-CoV-2 and Other Emerging Viruses”.

The manuscript number is viruses-1794154.

We have added in the material and methods section the information required by reviewer 2

Sincerely Yours,

Olivier Preynat-Seauve

This manuscript is a resubmission of an earlier submission. The following is a list of the peer review reports and author responses from that submission.

Round 1

Reviewer 1 Report

General comment

This study by Abdul et.al. developed a faster, novel and cost-effective full-length Spike/ACE2-dependant cell-to-cell fusion assay, based on a split luciferase reporter to measure fusion-inhibitory antibodies in serum of infected/vaccinated individuals. The methods are described in detail and manuscript is well written. However, I have a few concerns related to results and their representation.

Specific comments

1.   The main goal of the developed assay is to measure the fusion inhibition by serum Abs. however, the fusion directed antibodies are dominantly non-neutralizing as compared to RBD specific Abs. Also, the fusion directed mAbs are broadly neutralizing but they exhibit poor potency. Their frequency is very rare, but might improve post booster dose. It would great if authors can discuss this as a limitation of the assay and revise the manuscript text/results accordingly.

2.     I am confused why there is a variation in luminescence measurement in serial dilutions e.g. Fig 2C, patients between 20 and 25 at 1/512 dilution are showing higher luminescence as compared to 1/8 dilution. It would be better to show an x-y plot or raw values as a supplementary tables for each point shown in the main figures in Fig 2 to Fig 5.

3.     It would be a more clear representation in Fig. 2-5, if %inhibition is shown for the representation of the results in place of luminescence%. Luminescence % is confusing.

Reviewer 2 Report

In the manuscript “A cellular assay for spike/ACE2 fusion: quantification of fusion-inhibitory antibodies after COVID-19 and vaccination” by Abdul et al the authors report a new, cheap, well standardized for automation method based on split luciferase for detection of neutralizing antibodies in the serum of individuals.

The MS reports an interesting study, where a lot of standardization has been made to make the use of a simple method for evaluation of neutralizing Ab in the serum. The authors used 30 seronegative samples and implemented an internal control to reduce operatorʾs variability to determine the cut off values. When testing the samples from Covid-19 patients the majority of samples were detected as positive for the presence of neutralizing Ab. Additionally, very interesting are the results on case-reports showing the individuality of the immune response to mRNA vaccination.

Major comment:

1. Although I agree that already available kits have some shortcomings and that e.g. anti-RBD ELISA can not discriminate between Ab and neutralizing Ab, the authors should compare their luciferase assay with commercial one to really claim that commercial kits e.g. fail to detect neutralizing Ab at low levels, but their method can.

Minor comment:

1. Claim that split luciferase is a new method is misleading. There have been already publications published showing the use of split b-galactosidase (Theuerkauf SA et al., iScience, 2021) and split luciferase (Plaper et al., Science Reports, 2021; Manček-Keber et al., FASEB J, 2021) for measuring syncytia formation through spike and ACE2 interaction.